# Using Historical Responses to Shoreline Change on Australia's Gold Coast to Estimate Costs of Coastal Adaptation to Sea Level Rise

**Daniel Ware** [1],*, **Andrew Buckwell** [2], **Rodger Tomlinson** [1], **Kerrie Foxwell-Norton** [3] **and Neil Lazarow** [4]

1   Griffith Centre for Coastal Management, Griffith University, Gold Coast Campus,
    Gold Coast, QLD 4222, Australia; r.tomlinson@griffith.edu.au
2   Griffith Business School, Griffith University, Nathan Campus, Nathan, QLD 4222, Australia;
    a.buckwell@griffith.edu.au
3   Griffith Centre for Social and Cultural Research, Griffith University, Gold Coast Campus,
    Gold Coast, QLD 4222, Australia; k.foxwell@griffith.edu.au
4   CSIRO, Land and Water, Canberra, ACT 2601, Australia; Neil.Lazarow@csiro.au
*   Correspondence: d.ware@griffith.edu.au; Tel.: +61-755528389

**Abstract:** Climate change impacts, sea level rise, and changes to the frequency and intensity of storms, in particular, are projected to increase the coastal land and assets exposed to coastal erosion. The selection of appropriate adaptation strategies requires an understanding of the costs and how such costs will vary by the magnitude and timing of climate change impacts. By drawing comparisons between past events and climate change projections, it is possible to use experience of the way societies have responded to changes to coastal erosion to inform the costs and selection of adaptation strategies at the coastal settlement scale. The experience of implementing a coastal protection strategy for the Gold Coast's southern beaches between 1964 and 1999 is compiled into a database of the timing, units, and cost of coastal protection works. Records of the change to shoreline position and characteristics of local beaches are analysed through the Bruun model to determine the implied sea level rise at the time each of the projects was completed. Finally, an economic model updates the project costs for the point in the future based on the projected timing of sea level rise and calculates a net present value (NPV) for implementing a protection strategy, per km, of sandy beach shoreline against each of the four representative concentration pathways (RCP) of the Intergovernmental Panel on Climate Change (IPCC) to 2100. A key finding of our study is the significant step-up in expected costs of implementing coastal protection between RCP 2.6 and RCP 8.5—from $573,792/km to $1.7 million/km, or a factor of nearly 3, using a social discount rate of 3%. This step-up is by a factor of more than 6 at a social discount rate of 1%. This step-up in projected costs should be of particular interest to agencies responsible for funding and building coastal defences.

**Keywords:** erosion; shoreline change; climate change; sea level rise; adaptation; finance; coastal protection; cost estimates

## 1. Introduction

Climate change impacts, sea level rise, and changes to the frequency and intensity of storms, in particular, are projected to increase the coastal land and assets exposed to coastal erosion. The selection of appropriate adaptation strategies requires an understanding of the costs and how such costs will vary by the magnitude and timing of climate change impacts. By drawing comparisons between past events and climate change projections it is possible to use experience of the way societies have

responded to changes to coastal erosion to inform the costs and selection of adaptation strategies at the coastal settlement scale.

The Bruun model [1] is an empirical relationship between sea level and shoreline position, which considers geomorphic characteristics of a beach to predict how the position of the shoreline will respond to sea level rise. The Bruun model provides a way of comparing past events of the landward movement of shorelines to climate change projections by inputting data on past events and using it to solve for sea level rise, rather than the conventional approach to solve for changes to shoreline position. The Intergovernmental Panel on Climate Change (IPCC) recognises the outputs of the Bruun model for adaptation planning, despite criticism of its capacity to predict the response of beaches to sea level rise [2].

The Gold Coast is a coastal city in eastern Australia that developed from a series of small agriculture-based towns to a large tourist-, construction-, and services-based economy during the late 20th century. Between 1964 and 1999, the southern beaches of the Gold Coast experienced significant changes to the area of land exposed to coastal erosion. In response to these changes, the local and state governments implemented a protection strategy through a series of large coastal protection projects, including the construction of seawalls, groynes, and breach nourishment.

The past experience of implementing a coastal protection strategy for the Gold Coast's southern beaches between 1964 and 1999 is compiled into a database of the timing, units, and cost of coastal protection works. Records of the change to shoreline position and characteristics of local beaches are analysed through the Bruun model to determine the implied sea level rise at the time each of the projects was completed. Finally, an economic model updates the project costs for the point in the future based on the projected timing of sea level rise and calculates a net present value (NPV) for implementing a protection strategy, per km, of sandy beach shoreline against each of the IPCC's four representative concentration pathways (RCP) to 2100.

## 2. Literature Review

Beaches exist in a state of dynamic equilibrium, where wind, waves, sediment supply, and human actions interact to determine the position over time. Erosion is the process that shapes the shoreline and can be distinguished temporally from short-term responses to storm events, beach rotation, where the alignment of a shore shifts in response to seasonal changes in wind and waves, and recession, a longer-term process where shorelines shift landward. A recent analysis of satellite data has shown that around one-quarter of the world's beaches were receding during the 33 years to 2016 [3]. In its Fifth Assessment Report (AR5), the IPCC states there is high confidence that erosion hazards will increase for coastal settlements [2].

Between 1902 and 2015, global mean sea level has risen by 0.16 m, with the rate of rise increasing from 1.4 mm per year between 1901–1990, to 3.6 mm per year between 2006 and 2015 [4]. Despite the existence of detailed historical shoreline records for numerous locations around the globe [5], attribution of changes to sea level rise has been possible only in limited locations, as attribution requires a sediment budget to describe and quantify all sources and sinks of sediment. This task is subject to constraints on data collection and data quality, which in practice force assumptions that create uncertainty at a scale that is likely to overwhelm the sea level rise signal [6]. In addition, beaches are not purely physical places; they also have a social dimension and, as many beaches are in close proximity to high-value and intensive development, management activities, such as beach nourishment and dredging, must also be considered.

Future projections for global mean sea level rise over the course of the century range from less than 0.43 m for RCP 2.6 to 0.84 m for RCP 8.5 [4]. RCP 2.6 is predicated on effective global efforts to reduce greenhouse gas (GHG) with emissions peaking around 2020 and consistently falling through the course of the current century to limit warming to 2 °C (degrees Celsius) above pre-industrial levels [7]. RCP 8.5, by contrast, is the continuation of our current emissions trajectory, where there is little curbing of GHG emissions, which would entail an increase of around 4 °C by the end of the century [7]. Despite

the agreement of the world's governments through the United Nations Framework Convention on Climate Change (UNFCCC) to limit warming to 1.5 °C at the 21st Conference of Parties (COP) in Paris in 2015, higher temperatures and flow-on consequences for sea level rise demand consideration in future coastal management decision making. In a report to the United States National Oceanic and Atmospheric Administration, Sweet et al. [8] recommend consideration of between 0.3 m and 2.5 m increase in sea levels by 2100.

The coast is home to a significant proportion of the human population. Depending on how far inland the boundary is set, it could be home to 23% of the global population, when set at 100 km [9], or over 50% of the population at 200 km, and 66% within 400 km [10]. Relative to global populations, the population of the coast would seem to be growing far more rapidly. Between 1990 and 2002. The population living within 100 km of the world's coast grew from 1.2 to 2.5 billion—more than 56%—when over the same period the world's total population increased by 14% [11]. Blackburn et al. [12] found that 16 of the world's 23 megacities are located within the coastal area. Neumann et al. [13] projected the global low-elevation coastal zone (LECZ) population could rise from 625 million in 2000 to more than a billion people by 2060. A global assessment of the potential impacts of sea level rise as a result of erosion, through application of the Bruun model, showed that, without adaptation, land area of between 6000 and 17,000 km$^2$ may be lost to erosion by the end of the current century. This may result in 1.6–5.3 million people being forced to migrate, with an associated migration cost of USD 300–1000 billion (not discounted) [14].

Adaptation is defined by the IPCC [7] as the process of adjustment to actual or expected climate and its effects, in order to moderate harm or exploit beneficial opportunities. From the first IPCC report in 1990 [15] to the most recent in 2014 [3], adaptation for coastal settlements has been framed as a selection between three strategies: retreat, accommodate, and protect:

- Retreat involves no intention to protect the land from the sea. The coastal zone is progressively abandoned, and ecosystems enabled to shift landward. This approach can be motivated by the associated excessive economic or environmental impacts of protection. In the extreme case, an entire area may be abandoned, following a significant event.
- Accommodation implies that people continue to use the land at risk but do not attempt to prevent the land from being flooded. This option includes erecting emergency flood shelters, elevating buildings on piles, converting agriculture to fish farming, or growing flood- or salt-tolerant crops.
- Protection involves hard structures such as seawalls and dikes, as well as soft solutions such as dunes and vegetation, to protect the land from the sea so that existing land uses can continue (Gilbert and Vellinga [15]).

Initially, the process of adaptation was described as a single decision for a settlement to select between one of the three options [15]. In more recent years, the concept of 'adaptation pathways' has emerged, recognising that appropriate responses depend on the magnitude and timing of impacts, and seek to inform the identification of responses to such changes as an ongoing process rather than a single decision. Adopting an adaptation pathways approach involves the development of a strategic vision and changes to institutional arrangements that enable experimentation and learning so that choices along pathways can be altered in response to predefined triggers [16]. The implication is that for a coastal settlement all three strategies (retreat, accommodate, and protect) might be adopted in combination, depending on the rate of sea level rise or the impacts of coastal hazards. The important aspect here is that adaptation is rarely a specific, planned, and rationally-determined decision, but rather a change in the context within which decisions are made through altering the rules, values, or knowledge [17].

While information on future climate is important in motivating coastal governance arrangements to identify appropriate approaches from the three board strategies, it is not sufficient. Selecting from one of the three strategic options requires consideration of a range of other factors, including trade-offs between social, cultural, economic, and environmental values, future development scenarios,

institutional arrangements, and costs of implementation. The uncertainty associated with the timescales under consideration, both in terms of climate change impacts and the social and economic circumstances of the settlements that will be subject to these changes, requires the use of risk management and scenario testing, in addition to the application of established policy analysis tools, such as cost benefit analysis.

Based on a synthesis of experiences in coastal zone management, Tobey et al. [18] identify technical effectiveness, costs, benefits, and implementation considerations as the four key considerations for selecting adaptation approaches for coastal areas. Recent developments in adaptation theory would indicate that timing is an important fifth consideration—when to act, how long to continue an action, in what sequence to carry out those actions, and when an action should be altered [19].

Within adaptation research, time is highly relevant, due to the nature of the problem (sea level rise) being either present or future [20–22]. However, the framing of processes, pathways, and transitions suggest that there are opportunities for a greater contribution from the discipline of history. Both Adamson et al. [22] and Parsons and Nalau [21] explain the limited integration of history within adaptation research as a result of the disavowal of environmental determinism within the academic field of history, resulting in other disciplinary perspectives filling the gap and leading to accusations of over-simplification and neo-determinism. Despite this, both papers argue for the potential of historical research to contribute to the development of adaptation theory, particularly through empirical studies that uncover societal relations to climate in a particular place, while balancing impact attribution to climate change with power relations, social structures, technologies, economies, beliefs, values, and narratives.

Unsurprisingly, the discipline most concerned with past shoreline change has been geomorphology, with numerous studies examining the extent and mechanisms behind past shoreline change from recorded measurements. Houston and Dean [23] used shoreline position records for the Florida Coast from the 1850s to 2000s to identify how the combination of waves, onshore transport, and sea level rise are responsible for areas of accretion. Alberico et al. [24] used maps to derive shoreline position for southern Italy for nine time steps between 1870 and 1990 and identified increasing stability from the past to present. There are also a range of studies that make use of remote sensing, aerial imagery [25], and multispectral data [2] to identify shoreline changes over shorter time periods, from the 1950s for imagery and the 1980s for multispectral data, but across larger spatial scales, including globally [2].

Paleogeomorphology has also sought to define the extent and mechanisms behind past shoreline change through the reconstruction of records, using shipping logs from the 1800s to 2000s to reconstruct periods of greater erosion and show a relationship to Interdecadal Pacific Oscillation for the east coast of Australia [26]. Anderson et al. [27] use radiocarbon dating of artefacts of past human coastal settlement in Fiji to identify periods of stability and dynamism in the position of a coastal dune system from 2700 years before the present. This final example is unusual in the consideration of human responses to shoreline change, as most other examples focus solely on biophysical drivers and responses, or where human activity is recognised as affecting shoreline change, rather than being affected by shoreline change.

Histories of coastal settlement tend to focus on land use and development of coastal areas and characterise the drivers for these in terms of access to resources, in particular food and fresh water, and of maritime transport and the trade opportunities provided by proximity to the ocean [28]. In a history of human coastal settlement, which traces the migration of humans from Africa across the globe, Gillis [28] develops an alternative to the agricultural savannah theory of human civilisation, based on the connections between human civilisation and global migration and the coast. Hoskins [29] traces the history of human occupation of the New South Wales (NSW) coast in Australia from Indigenous Peoples first settlement (circa 65,000 years ago) through European invasion, to the present, highlighting the relatively recent trend towards the recreationalist usage of the coast, as epitomised by the contemporary Gold Coast. Salter [30] documents the history of Stradbroke Island, off the east coast of Australia, and its different uses and settlement patterns over time, with particular recognition of the shifting shorelines changing connections to the mainland and the loss of settlements and communities

as a result of shoreline change. While the majority of treatments of the history of coastal development deal with shoreline change, it is in a largely qualitative sense and, in the limited instances where climate change is projected, there is limited emphasis on the connections between past responses to shoreline change and future projections.

The use of the discipline of history in the forecasting of social and environmental responses to climate change was initially proposed by Glantz [31] through the framework of using 'analogies' as a process of reasoning, where known facts about a base case are used to make predictions about yet-to-be-observed relationships in a target problem. There are two primary criticisms of this approach. The first is that climate change is altering the Earth's climate with a speed and magnitude that has not been previously experienced [32]; and the second is that, as societies are changing so rapidly (in terms of technology, population, and connectivity), the way they may respond to climate change will be incommensurable to the way past societies have responded to climate or environmental changes [33]. In response, there are two primary arguments in favour of the application of temporal analogues to climate change. Firstly, these criticisms fall into the same trap as environmental determinism in overestimating the role of climate in social change, rather than focusing on the social structures as responses [21]; the second is that, despite differences in the social and environmental conditions, analogues provide significant empirically-based insights into how society experiences and responds to change [31,34].

With the exception of Hoagland et al. [35], examples of the application of climate change analogue approaches, as described by Ford et al. [34] and Glantz [31], to shoreline change to forecast climate change experiences and responses cannot be identified. Hoagland et al. [35] used a dataset of historical beach nourishment projects comprising over 400 individual nourishments between 1930 and 2011 to develop a statistical relationship between site characteristics and overall project cost to enable more accurate estimates of the cost of beach nourishment projects as a result of climate change.

Time is a central dimension to climate change research. However, the emphasis in the literature and political discourse is on projecting from the present into the future. The orientation towards the future is driven by our attempts to understand the social, cultural, environmental, and economic consequences of increasingly accurate projections for future climate change provided by rapidly improving capacity for climate system modelling. Adaptation is increasingly being understood as a process of societal change. Within this framing, we argue there are significant opportunities to learn from how past societies have changed in response to climatic and environmental processes, to make predictions how they will respond in the future.

## 3. Gold Coast Coastal Erosion between 1960s to 2000s

The southern Gold Coast in Australia provides an opportunity to study experiences of and responses to the effects of multi-decadal shoreline change. The Gold Coast is a large coastal strip city with a population of 576,918 residents, an annual growth rate of around 2.3% since 2011, and a projected population of 798,000 by 2031. The city is bounded by the border with the state of NSW to the south, the Logan river to the north, forested Mt Tambourine and the Springbrook Plateau to the west, and the Pacific Ocean to the east. The climate is sub-tropical, with rainfall patterns and sudden river flooding influenced by El Niño/La Niña cycles. It is generally outside tropical cyclone range; however, remnant tropical cyclones have been responsible for significant weather events for the city. The Gold Coast is also bound by the City of Gold Coast Local Government area. There are 55 km of beaches and 270 km of waterways—natural and trained. European invasion and occupation of the area commenced with initial settlements at Southport in 1874 and Coolangatta in 1883, mainly associated with timber extraction activities. The city has developed rapidly through the latter part of the 20th century, particularly following World War II, during which time US and Australian soldiers fighting in the Pacific Campaign came to the region for rest and recreational leave, stimulating the area's tourism industry, which grew rapidly once restrictions on access to construction materials were lifted at the end of the war. The development of the Gold Coast significantly reshaped the landscape; first with

sand mining and then with the draining and filling of wetlands for conversion into residential and tourism developments.

In Australia, the governance of coastal protection is a complicated matter. Australia is a federation of eight state and territory governments and almost 600 local governments, which are referred to variously as city, shire, and regional councils. Roles and responsibilities between state and the federal government is distributed on the basis of the Australian Constitution, which does not recognise local governments. Governance of natural hazards was not addressed by the Constitution and, as such, coastal protection remains a residual power of state governments [36]. As a result, the federal government, which retains the vast majority of tax-raising capability, has limited involvement in coastal protection [37,38], consisting of coordination of national disaster management and information provision. State governments set coastal management policy within their jurisdiction, while local governments are responsible for implementing land use planning, disaster management, and coastal management (including management of coastal hazards). The state governments have granted local governments the power to create local instruments with legal effect and local policies that apply to the shoreline [39].

In the summer of 1967, a series of storms severely damaged the beaches of the Gold Coast. The event was so significant the Australian Defence Force was mobilised to support recovery efforts. As a result, in 1968 the Queensland State Government became the first state in Australia to implement legislation designed to enable coastal planning and control development within areas exposed to coastal erosion and flooding hazards [40]. The following decades saw the Gold Coast City Council and the Queensland State Government deliver what were, at the time, some of the largest coastal protection works ever undertaken in Australia.

While, at the time, the cause of the damage was attributed to storm erosion, the scale of the damage and the establishment of government intuitions promoted a significant period of knowledge acquisition in relation to local coastal processes. This process was eventually able to attribute the damage from the 1967 and subsequent storms to the trapping of sand to the south by an earlier extension of the Tweed River training walls by 400 m, in 1962, by the New South Wales (NSW) Government (see Figure 1).

Shoreline change and the distinctive features of Gold Coast beaches are significantly influenced by longshore transport. Longshore transport is the movement of sand parallel with the shoreline as a result of energy from wind and waves. The rate of longshore transport is primarily a function of the orientation and energy of prevailing wind and waves relative to the shoreline. For a given energy, the closer to parallel the prevailing wind and wave direction is to the shoreline, the greater the rate of longshore transport. For the Gold Coast, the prevailing wind and wave direction is from the southeast and the shoreline faces east, which results in an annual longshore transport rate of approximately 500,000 $m^3$ [41].

Despite awareness of the phenomena of longshore transport in the 1960s, the reliance of Gold Coast beaches on the flow of sand northwards from NSW was not fully understood by either state government at the time the training walls were extended. Even when the Queensland State Government commissioned Delft Hydraulics Laboratory in The Netherlands to investigate the causes and responses to beach erosion, the findings assumed there was limited transport of sand from NSW into Queensland beaches [42]. By the 1980s, however, sufficient data had been collected to start indicating the potential for sand transport from NSW northwards onto Gold Coast beaches, and the visual evidence of the volumes of sand accumulating at the Tweed River's southern training wall were a clear indication of what was occurring. By 1967, the extended training walls were estimated to have trapped more than 2 million $m^3$ of sand, which otherwise would have been taken north onto Gold Coast beaches. Thus, while the storms in 1967 were the proximate cause of the severe damage to the Gold Coast beaches, had the sand been transported to Coolangatta, rather than being held, accumulating at Letitia spit, the damage to the north would have been reduced.

Boak et al. [43] documented the events that resulted from the Tweed River training wall extension and the coastal protection works constructed in response on the Southern Gold Coast between 1960

and 2001. Castelle et al. [44] documented beach nourishment works, volumes, and sand placements on the southern Gold Coast from the 1980s to 2000s in response to the reduction in longshore transport. Jackson et al. [45] reviewed the beach nourishments on the Gold Coast from the 1970s to 2013 to estimate the total volume of sand added to Gold Coast beaches and identified how the placements and sources have evolved. Strauss et al. [46] analysed data on shoreline position from the 1970s to 2000s; however, the influence of human activities, in the form of beach nourishment, altered the shoreline position in such a way as to obscure the full impact of the extension of the training walls. We draw upon all these sources to reconstruct the coastal management works undertaken during this time.

In 1994, the NSW and Queensland State Governments finally reached an agreement to re-establish the flow of sand from NSW into the Gold Coast beaches and in 1999 construction commenced on the permanent Tweed River Entrance Sand Bypass and Jetty to the south of the Tweed River mouth, which dredges sand trapped by the walls and pumps it across the river mouth, north, to be discharged on the north side of the river at Rainbow Bay to flow onto Gold Coast beaches.

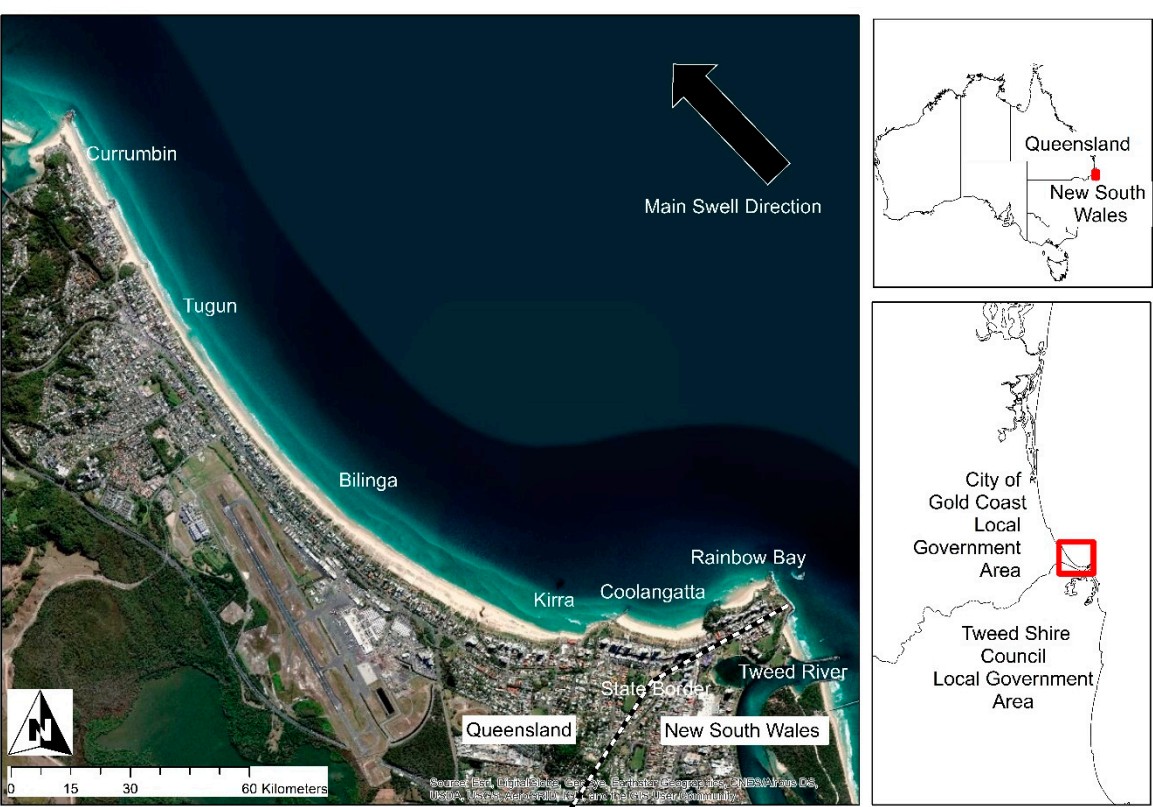

**Figure 1.** Map of Gold Coast showing key locations and main swell direction.

## 4. Method and Results

In previous sections we identified the well-established methods for the study of shoreline change. While there have been numerous attempts to project how climate change will impact on shoreline change, there is limited study of the use of past events and community responses to develop climate change analogues. We apply the following four steps to a case study of shoreline change at the southern Gold Coast in Queensland Australia, between the 1960s and 2000s:

1.　Identify a counterfactual against which to assess the change.
2.　Define a base case and quantify the change, experience, and response.
3.　Draw a relationship between the change and projected sea level rise under future climate change.
4.　Apply the knowledge of the base case to a problem case (future sea level rise) to develop an understanding of the experience and societal response to climate change.

The data for this paper was collected through interviews with individuals who held positions of responsibility or influence for planning and delivery of coastal protection at the southern Gold Coast through the period in question, either directly from within government, or indirectly as consultants, academics, or representatives of stakeholder groups. This interview data was supplemented through academic literature reviews, as well as historical records accessible through the coastal knowledge hub and the Gold Coast Historical Studies Library, which included reports from both the Queensland State Government and City of Gold Coast and relevant media articles from the time.

### 4.1. Identify a Counterfactual Against which to Assess the Change

The first stage in the development of an analogue is to confirm that a counterfactual exists and can be established. A counterfactual is an estimate of the circumstances that would have prevailed had a change, which forms the basis of the analogue, not occurred. The establishment of a counterfactual is essential to have some basis against which to compare the experience and responses to a change. The existence of a counterfactual enables the attribution of experiences and responses to the change under examination. The absence of a counterfactual would provide significant difficulty in forecasting a change.

Through the establishment of a coupled wave and shoreline evolution model for the Gold Coast, and by running it in hindcast for the period 1950 to 2012, Teakle et al. [47] state that the "most significant impact on the Gold Coast system during this period resulted from the construction of the Tweed River training walls in 1963" [45]. The volumes of sand trapped by the extension of the training walls has been estimated, at various point in time, to be 5.7 million m$^3$ by 1983 [48] and 7.2 million m$^3$ by 1988 [49].

The counterfactual scenario for the period 1960 to 2000 is that the Tweed river training walls are not extended and sand continues to flow from NSW onto Gold Coast beaches, thereby maintaining their resilience to erosion events and avoiding the need for significant intervention in coastal processes through that period.

### 4.2. Define the Base Case and Quantify the Change, Experience and Response

The base case is the system that has experienced change. This step involves documenting the change, experience, and response, and, then, through a comparison to the counterfactual established in step 1, quantifying the experience and response attributable to the change.

The base case under observation is the shoreline of the southern Gold Coast and the actions necessary to respond to a receding shoreline when a protection strategy is adopted. The actions taken are the construction of coastal protection works necessary to deal with the removal of sand from the southern Gold Coast beaches as a result of the extension of the Tweed River training walls in 1962. The specific area of study is determined by the area impacted by the extension of the training walls, which is defined by Teakle et al. [47] as the shoreline from Point Danger in the south, northwards (from resultant longshore drift) to Burleigh Head (see Figure 1). The period under consideration is between 1962 and 2000, when a permanent bypassing solution commenced operations.

Table 1 is a summary of all recorded actions in the case study period, identifying the year of implementation, the general type and location of intervention, and the unit of measurement. Over the period 1968 to 2000 (when the sand bypass was implemented), there were 43 interventions, installing nearly 12,000 m of sea wall and 1150 m of groynes and nourishing beaches with nearly 11 million m$^3$ of sand. These interventions are assumed to be sufficient to have successfully managed the shortfall of sand deposition during the period when the training walls blocked passage of sand northwards.

**Table 1.** Southern Gold Coast coastal protection works.

| Year | Type | Location | Length (m) | Volume (m$^3$) |
|---|---|---|---|---|
| 1968 | Wall | Coolangatta | 929 | |
| 1972 | Groyne | Kirra Point | 180 | |
| 1972 | Groyne | Currumbin South | 200 | |
| 1972 | Wall | Rainbow bay | 200 | |
| 1973 | Wall | Kirra (protect the Highway) | 912 | |
| 1974 | Groyne | Kirra (Miles Street) | 120 | |
| 1974/1975 | Nourishment | Kirra | | 1,000,000 |
| 1974/1975 | Wall | Kirra | 912 | |
| 1974/1975 | Wall | Kirra to North Kirra | 1400 | |
| 1970–75 | Private Walls | Southern Coast | 20 | |
| 1975 | Wall | Tugun | 30 | |
| 1975/1976 | Wall | North Kirra to Bilinga | 1700 | |
| 1976 | Groyne | Tallebudgera | 230 | |
| 1975–80 | Private Walls | Southern Gold Coast | 123 | |
| 1980 | Groyne | 11th Ave Palm Beach | 70 | |
| 1980 | Groyne | 21st Ave Palm Beach | 70 | |
| 1980 | Wall | Palm Beach | 100 | |
| 1981 | Groyne | Currumbin Creek North | 160 | |
| 1983 | Wall | Palm Beach | 20 | |
| 1980–85 | Private Walls | Southern Gold Coast | 1265 | |
| 1985 | Geotextile Groyne | North Kirra Surf Club | 120 | |
| 1985 | Nourishment | Kirra | | 315,000 |
| 1985 | Nourishment | Kirra | | 215,000 |
| 1985 | Wall | Tugun | 40 | |
| 1987 | Wall | Kirra | 650 | |
| 1987 | Wall | Palm beach | 20 | |
| 1988 | Nourishment | Kirra/Bilinga | | 1,500,000 |
| 1989 | Nourishment | Southern Gold Coast | | 3,200,000 |
| 1989 | Nourishment | Southern Gold Coast | | 395,000 |
| 1985–90 | Private Walls | Southern Gold Coast | 1522 | |
| 1990 | Wall | Bilinga | 1330 | |
| 1990/91 | Wall | Palm beach | 40 | |
| 1990/91 | Wall | Tugun | 70 | |
| 1993 | Wall | Palm Beach | 20 | |
| 1994 | Wall | Tugun | 20 | |
| 1990–95 | Private Walls | Southern Gold Coast | 1233 | |
| 1995 | Nourishment | Southern Gold Coast | | 2,300,000 |
| 1996 | Wall | Kirra | 60 | |
| 1997/97 | Wall | Palm Beach | 60 | |
| 1997 | Nourishment | Southern Gold Coast | | 800,000 |
| 1998 | Wall | Palm Beach | 50 | |
| 2000 | Nourishment | Southern Gold Coast | | 1,100,000 |
| 1995–2000 | Private Walls | Southern Gold Coast | 634 | |

*4.3. Draw a Relationship between the Change and Projected Future Climate Change*

In order to apply the knowledge regarding a base case to forecasting a problem case (either the experience or responses to climate change) it is necessary to draw a relationship between the change under examination for the base case and climate change. The relationship could be drawn in a number of ways by time, representative concentration pathway, or amount of change. For the example case study, the first step is to make a connection between the experience of change and the climate change impact under examination.

Sea level rise is predicted to lead to a landward movement of the shoreline as beach profiles adjust to the increase in water levels by redistributing sediment from the dune to the sub-aerial profile of beaches (see Figure 2). Measurements of the profile of southern Gold Coast beaches have been recorded since the 1960s; however, they cannot be used to directly estimate the volume of sand removed from the system by the Tweed River training walls, as there has been significant interference from human

activities in the form of beach nourishment (see Table 1) [47]. Therefore, to relate the change that has occurred to an amount of sea level rise, we first need to establish the volume of sand removed over time (see Table 2).

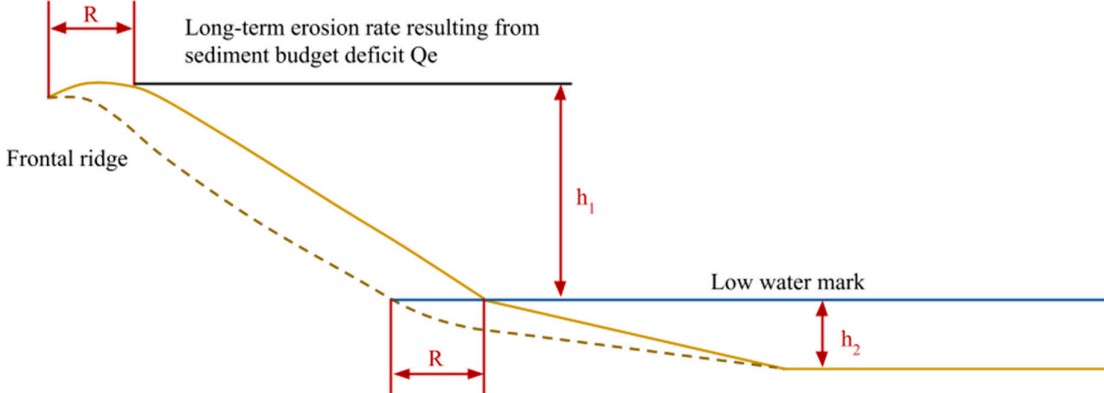

**Figure 2.** Relationship between sand volume loss and recession.

**Table 2.** Total sand volume lost from Gold Coast beaches, 1960–2000 (Source: [47]).

| Period | Total Cumulative Volume of Sand Withheld from Southern Gold Coast at Letitia Spit by Tweed River Training Walls (m³) |
|---|---|
| 1962–1965 | 153,000 |
| 1965–1970 | 613,000 |
| 1970–1975 | 2,111,000 |
| 1975–1980 | 3,711,000 |
| 1980–1985 | 5,089,000 |
| 1985–1990 | 5,923,000 |
| 1990–1995 | 6,502,000 |
| 1995–2000 | 7,234,000 |

To convert the volumes of sand removed to the amount of recession, a formula was provided by the Queensland Department of Environment and Heritage Protection (DEHP) [50], based on Figure 2 and Equation (1), below.

$$R = \frac{Qe}{(h1 + 0.5h2)} \tag{1}$$

where $Qe$ = erosion quantity in cubic metres per metre length of beach per year (m³/m/yr); $h1$ = height of frontal beach ridge above low water mark (metres); $h2$ = depth of closure (metres); and $R$ = long-term erosion rate (m/yr).

When Equation (1) is applied to the volume of sand lost from southern Gold Coast beaches from Table 2, the impact of nourishment and protection works on southern Gold Coast beaches can be removed to estimate the amount of recession that occurred at the southern Gold Coast between 1960 and 2000 in each (generally) five-year period between 1960 and 2000. This output is reported in Table 3.

Conversion of the shoreline recession into a sea level rise equivalent is achieved by using the model relationship between sea level rise and landward retreat of Bruun [1], reproduced below as Equation (2). The Bruun model is the most widely used method for predicting beach shoreline response to sea level rise in isolation of other factors, such as waves and sediment availability [51]. The Bruun model assumes that as sea level rises, a given beach's cross-sectional shape (profile), which is determined by sediment size and wave climate, will maintain its shape and rise with the sea level from sediment being supplied from the beach, causing it to recede (see Figure 3).

**Table 3.** Recession of southern Gold Coast beaches due to sand trapped by extended Tweed River training walls between 1960 and 2000.

| Period | Qe (m³) where Length of Shoreline Point Danger to Burleigh Heads = 13,000 m | R (m) where h1 = 3 and h2 = 12) |
|--------|------|------|
| 1962–1965 | 12 | 1 |
| 1965–1970 | 47 | 5 |
| 1970–1975 | 162 | 18 |
| 1975–1980 | 285 | 31 |
| 1980–1985 | 391 | 43 |
| 1985–1990 | 456 | 51 |
| 1990–1995 | 500 | 56 |
| 1995–2000 | 556 | 62 |

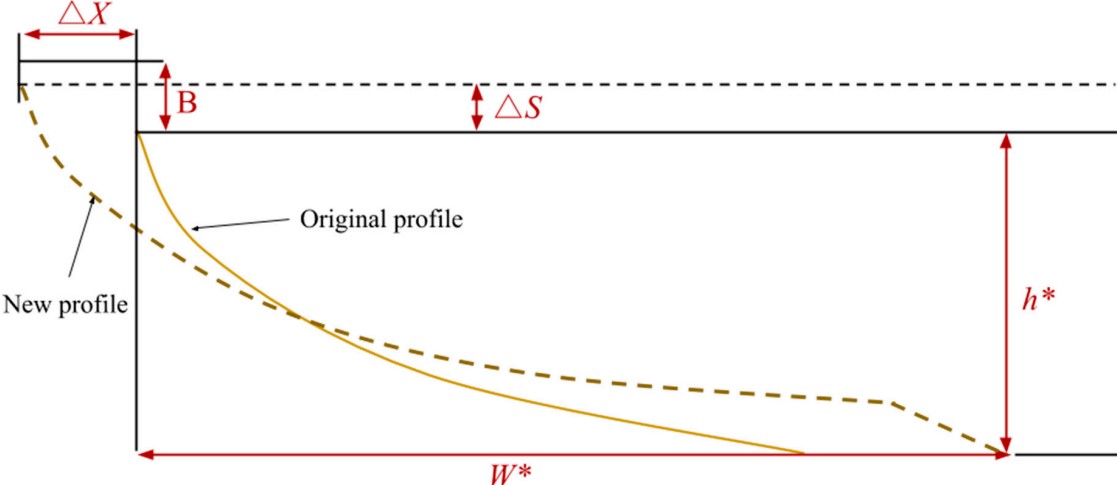

**Figure 3.** Beach response to sea level rise (Dean and Houston, 2016). $\Delta X$ = shoreline change; $\Delta S$ = sea level rise; B = beach berm height; $h$ = depth of closure; and $W$ = distance from berm to depth of closure.

The Bruun model has been shown to provide a general alignment with results of shoreline change for a given sea level rise in wave tank study environments [52,53], albeit without longshore transport and open coasts without inlets [54,55]. Limitations of the Bruun rule are based on three main issues: poor data quality, limited data period, and insufficient quantification of sediment budget [56]. These are valid criticisms, yet are a critique of the application of the model, rather than the model itself. While uncertainty remains regarding the quantification of changes to erosion hazard as a result of sea level rise, the evidence from the IPCC supports an increasing trend towards greater hazard [3].

$$R = \left(\frac{L}{B+h}\right) \times S \qquad (2)$$

where $R$ = landward retreat; $h$ = closure depth; $L$ = horizontal distance from shoreline to offshore position of $h$; $B$ = height of the frontal dune; and $S$ = sea level rise.

To solve for sea level rise, we rearrange Equation (2) as Equation (3) below.

$$S = R \times \left(\frac{B+h}{L}\right) \qquad (3)$$

The method described above shows that the amount of sand trapped by the 1962 extension of the Tweed River training walls would, without intervention, be expected to cause 62 m of recession of the shoreline across the southern Gold Coast between 1960 and 2000 (Table 4). The amount of recession

caused by the trapping of sand is the equivalent to the recession that would result from 1.85 m of sea level rise (Table 5).

**Table 4.** Implied shoreline retreat and implied sea level rise (closure depth (h) = 12 m; height of dune (B) = 3 m; length of beach between Snapper Rocks and Burleigh Head = 13,000 m; horizontal distance from shoreline to offshore position of closure (L) = 500 m).

| Year | Erosion Quantity ($m^3$) | Erosion ($m^3$/m) | Retreat (m) | Sea Level Rise Equivalent (m) |
|------|------|------|------|------|
| 1960 | 153,000 | 12 | 1.3 | 0.04 |
| 1965 | 613,000 | 47 | 5.2 | 0.16 |
| 1970 | 2,111,000 | 162 | 18.0 | 0.54 |
| 1975 | 3,711,000 | 285 | 31.7 | 0.95 |
| 1980 | 5,089,000 | 391 | 43.5 | 1.31 |
| 1985 | 5,923,000 | 456 | 50.6 | 1.52 |
| 1990 | 6,502,000 | 500 | 55.6 | 1.67 |
| 1995 | 7,234,000 | 556 | 61.8 | 1.86 |

**Table 5.** Sea level rise equivalent of recession of Southern Gold Coast Beaches due to sand trapped by extended Tweed River training walls, 1960–2000.

| Period | Sea Level Rise Equivalent (where H = 12, B = 3 and L = 500) |
|------|------|
| 1962–1965 | 0.04 |
| 1965–1970 | 0.16 |
| 1970–1975 | 0.54 |
| 1975–1980 | 0.95 |
| 1980–1985 | 1.31 |
| 1985–1990 | 1.52 |
| 1990–1995 | 1.67 |
| 1995–2000 | 1.86 |

*4.4. Apply the Knowledge of the Base Case to a Problem Case to Develop Understanding of the Experience and Societal Response to Climate Change (Results)*

Through the previous sections we have established both the responses to shoreline change for the base case (Table 4) and established a comparison between the shoreline change that occurred and what would be the equivalent sea level rise to cause such a change, based on the Bruun model (Table 5). This base case, therefore, represents a series of actual historical defensive responses to stabilise the shoreline for sea level rise adaptation in response to a series of erosion events (in contrast to pursuing retreat or accommodate responses).

Typical approaches to coastal hazard adaptation tend to consider packages of planned, strategic, defined approaches, based on rational planning, cost benefit analysis, and a trust that responsible agencies carefully and pre-emptively carry out any coastal protection [57–60]. Our base case suggests that, in practice, responses to coastal hazard—in this instance, an implied sea level rise—are reactive, often localised in scale, planned and funded by a mixture of public and private sources, and are implemented on demand, often in response to a shortfall in sand or a specific erosion event. By estimating the costs of implementation of the coastal protection works historically carried out in the base case, and then aligning the costs of protection with the implied sea level rise, it is possible to calculate how the costs of a 'defend' approach transpire over time in a manner similar to that implied by the 'adaptation pathways' concept, rather than as a single project, as is often the case with plans for adapting coastal protection to sea level rise.

The output of our approach is an economic model that estimates the future value cost, per kilometre, of a defensive approach for sea level rise as projected in the IPCC's four representative concentration

pathways (RCP 2.6, 4.5, 6.0, and 8.5) in its Fifth Assessment Report [7]. A summary of our methodology is shown below; the details of the economic model methodology are set out in Supplementary Information.

Economic Model

The approach taken in economic modelling was as follows:

1. Establish a historic timeline of projects implemented in response to erosion cause by shortfall of sand on Gold Coast beaches on a 13 km stretch of coast that was impacted by the sand deficit as a result of the construction of the Tweed River training walls.
2. Estimate current costs of historic responses, based on literature review of projects of similar, contemporary projects.
3. Estimate the implied sea level rise at the time of each project's implementation. Sea level rise is always estimated relative to the IPCC 2005 baseline [61].
4. For each project, we determined the years from present over which the implied sea level rise would occur, based on the projections for the four RCPs for the Gold Coast [61].
5. Inflate the unit costs for the project types (sea walls, groynes, and beach nourishment) to the year 2100. Apply a discount rate (r) to future project costs, based on the years from the present that the implied sea level rise would occur.
6. Apply an inflating maintenance cost for seawalls and groynes (no maintenance costs are associated with beach nourishment projects).

The results of our model are reported in Table 6, which shows the indicative costs of coastal protection attributable to sea level rises (SLR) associated with the IPCC RCP scenarios for a predominantly beached, urban coastal area, such as the southern Gold Coast. The net present value (NPV) cost, using a 3% discount rate (r), of a defensive approach to sea level rise, per kilometre of coastal area, associated with RCP 2.6 to 2100 (from 2020) is $574,000/km; for RCP 8.6, this value inflates to $1.715 million/km.

**Table 6.** Costs of coastal protection between 2020 and 2100 (2016 USD) (r = 3%; inflation = 1.7%).

| RCP | Computed SLR (m) | Total Future Value (FV) | NPV | FV/km | NPV/km |
|---|---|---|---|---|---|
| 2.6 | 0.615 | $31,937,000 | $7,460,000 | $2,457,000 | $574,000 |
| 4.5 | 0.770 | $63,364,000 | $12,186,000 | $4,874,000 | $937,000 |
| 6 | 0.780 | $63,865,000 | $12,063,000 | $4,913,000 | $928,000 |
| 8.5 | 1.120 | $346,821,000 | $22,290,000 | $26,679,000 | $1,715,000 |

We also ran a sensitivity analysis using discount rates of 1% (a 'social' discount rate) and 7.5% (the rate recommended by the Australian government for assessing infrastructure projects). These results are reported in Tables 7 and 8 and are depicted in Figure 4.

**Table 7.** Costs of coastal protection between 2020 and 2100 (2016 USD) (r = 1%; inflation = 1.7%).

| RCP | Computed SLR (m) | Total Future Value | NPV | FV/km | NPV/km |
|---|---|---|---|---|---|
| 2.6 | 0.615 | $37,771,000 | $23,150,000 | $2,905,000 | $1,780,800 |
| 4.5 | 0.770 | $70,941,00 | $40,271,000 | $5,457,000 | $3,098,000 |
| 6 | 0.780 | $71,339,000 | $40,282,000 | $5,48,000 | $3,099,000 |
| 8.5 | 1.120 | $369,836,000 | $145,509,000 | $28,449,000 | $11,193,000 |

**Table 8.** Costs of coastal protection between 2020 and 2100 (2016 USD) (r = 7.5%; inflation = 1.7%).

| RCP | Computed SLR (m) | Total Future Value | NPV | FV/km | NPV/km |
|-----|-----|-----|-----|-----|-----|
| 2.6 | 0.615 | $25,027,000 | $713,000 | $1,1925,000 | $55,000 |
| 4.5 | 0.770 | $54,759,000 | $1,215,000 | $4,212,000 | $93,000 |
| 6 | 0.780 | $55,247,000 | $1,158,000 | $4,250,000 | $89,000 |
| 8.5 | 1.120 | $316,189,000 | $502,000 | $24,322,000 | $39,000 |

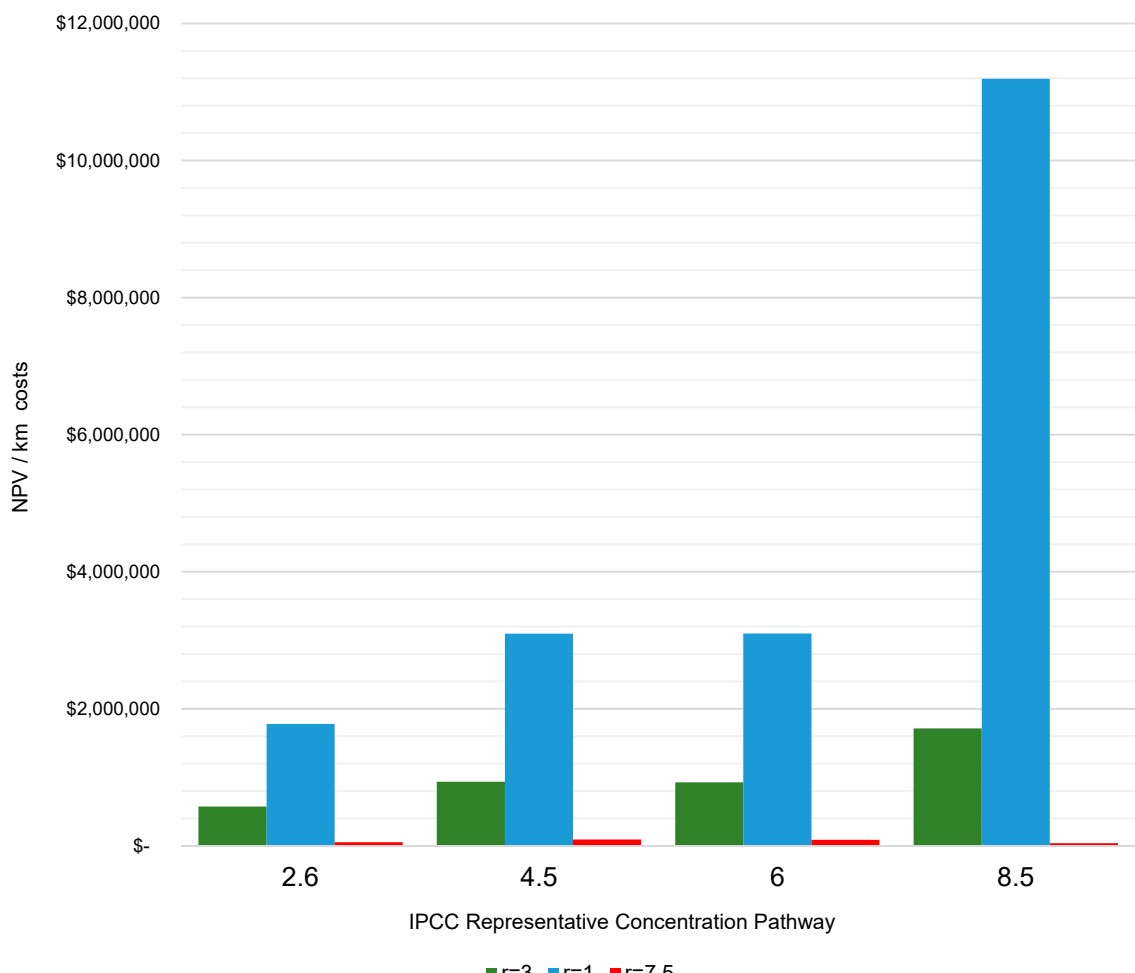

**Figure 4.** Derived net present value/km of coastline defensive adaptation approach to climate change adaptation for the southern Gold Coast.

## 5. Discussion and Conclusion

We estimated indicative costs to implement a coastal protection strategy for climate change adaptation involving a mix of constructing sea walls, groynes, and nourishing beaches with sand for each of the IPCC's RCPs at a predominantly beached, urban location. These costs are representative of distributed, generally reactive, and adaptive decision-making, considered when threats become apparent, at a highly local scale. We assume that the costs associated with and the process of future decision making for coastal protection are likely to broadly align with how decisions have been made in the past. The hazard (and subsequent continuation of a worsening of that hazard) associated with loss of sand on the beaches to the north of the Tweed River training walls emerged as new knowledge of coastal processes came to light, in much the same way knowledge of the hazards associated with sea level rise as a result of climate change has emerged in the last 30 years. Whilst our method makes no attempt to calculate and compare costs to accumulated benefits (from avoided damage costs or damage

curves, for example), we can assume that each separate historical defensive action was implemented at least through an *implicit* cost benefit analysis; that is, the benefits that were likely to accrue to each project were perceived to be greater than the costs at that given time.

A key finding of our study is the significant step-up in expected costs of coastal defences between RCP 2.6 and RCP 8.5—from $574,000/km to $1.7 million/km (r = 3%), or a factor of nearly 3. This step-up is by a factor of more than 6 at a social discount rate (r = 1%). This step-up in projected costs should be of particular interest to agencies responsible for coordinating, funding, and building coastal defences (in Australia, this has typically been local government; in this instance, the Gold Coast City Council). As global climate policy negotiations continue to lag behind stated ambitions and global emissions of greenhouse gases continue to rise [62], the costs of implementing coastal protection with higher emission concentrations are significantly higher with RCP 8.5 than with other scenarios.

A second, more subjective, finding is the apparent modest defend costs in our results. Beach-front 'unimproved' private property prices in the affected areas are between AUD 2500 and AUD 4000 per $m^2$, making them some of the most expensive properties in the state of Queensland, outside the two major centres' central business districts. Crude valuation of defensive costs of a 200 m stretch of coastal development in the suburb of Kirra, for example, has an NPV of AUD 3562, which defends private property (land only, excluding any buildings) to the current value of approaching AUD 50 million in directly exposed property. From a coastal adaptation perspective, this suggests costs of defensive actions (a combination of sea walls, groynes, and beach nourishment) will never approach replacement cost damages. Assessing damage curve values for affected properties, as an avoided cost, i.e., benefit of coastal protection, is beyond the scope of this paper; however, it would be a useful adjunct to our work, as, in this location, property values are so high.

We should remark that the authors do not advocate adaptation to sea level rise in the coastal zone through a protection strategy undertaken in an ad hoc, reactive way. Rather that, based on past responses to an implied sea level rise, in a complex, multi-stakeholder environment, responses are subject to the uncertainty inherent in the RCP scenarios and the politics of competing private and public interests. If we accept this likely scenario, then our best estimate of defensive costs is one that reflects what has happened in the past, rather than one equating to the rational ideal of Park et al. [63], where small-scale incremental adaptation actions eventually become elevated to transformational and strategic coordinated adaptation. Cooper and Lemckert [64] note that unplanned response to sea level rise is likely "characterised by periodic crises, growing uncertainty and public unease and would have marginal chances of success", but nevertheless the City of Gold Coast would "probably" survive sea level rise of 1 m [64]. Often studies on the benefits and costs associated with coastal adaptation [65,66] are based on assumptions on a range of environmental conditions and economic parameters over costs associated with adaptation into the long-term future—they are thus subject to "deep uncertainty" [67,68], as far future benefits are hard to quantify [69]. Therefore, typical tools of assessment, such as cost benefit analysis (CBA), can provide a false sense of certainty for something that is inherently deeply uncertain. CBA is based on a 'predict-then-act' mindset, which is rooted in the 'expected utility' hypothesis of classical decision theory, where there is a "tendency to view model outputs as objective, capable of defining optimal goals and strategies for which climate policy should strive, rather than as exploratory tools within a broader policy development process" [68]. Such an approach assumes decision makers can make reasonable predictions about the future—or at least reliably characterise the probabilities of different outcomes. However, climate change, extreme weather events, and private, social, and institutional responses to the impacts of climate change are, by definition, unprecedented and, therefore, unpredictable. Furthermore, our efforts to contain or control for uncertainty in the future may only serve to further complicate and confuse responses. Historical observations of both social and scientific data provide an important counterbalance for rational planning.

Responding to increased coastal hazard, as a result of sea level rise induced by anthropogenic climate change, is one of the key adaptation challenges [70]. For intensely developed areas, such as

Australia's Gold Coast, which have significant private and public infrastructure investment in the coastal hazard zone and limited unmodified habitats, options for adaptation are often limited to engineered options, such as sea walls, sea dikes, and beach nourishment. Where habitat conditions allow for nature- or ecosystem-based adaptations, the significant land and building values ensure the more expensive, but also more focused, singular defensive strategies still provide net benefits, at least under scenarios of more limited sea level rise [54]. This adaptation will be funded through a combination of both private and public funding on a spectrum of planned, strategic adaptations, to small-scale, responsive, retrofitted assets. The relative burden between the two is dependent on a range of specific local issues; however, it is likely that much of the burden, in Australia, will fall to local governments. Despite the relatively modest cost we have estimated, the costs of coastal adaptation are predicted to stretch local government revenue-raising abilities [71]. However, Australian local government areas have favourable asset-to-liability ratios and low levels of debt, and can potentially borrow against a robust revenue base (rate payers and businesses). Therefore, financing options may provide a useful pathway to closing the gap between likely costs and infrastructure budgets. Gaining insight into these likely costs (both public and private) will be vital in planning in the forthcoming years.

## 6. Limitations

The analysis assumes a consistent beach profile for the 13 km of shoreline from Snapper rocks to Burleigh headland, however, some variation in the profile should be expected. This is unlikely to significantly impact the results expressed as per/km given profile variation is typical of most sections of exposed coast.

**Supplementary Materials:** The following are available online at http://www.mdpi.com/2077-1312/8/6/380/s1, included with this manuscript submission.

**Author Contributions:** Conceptualization, D.W.; methodology, D.W.; formal analysis, D.W. and A.B.; investigation, D.W.; resources, R.T.; writing—original draft preparation, D.W.; writing—review and editing, A.B., R.T., K.F.-N. and N.L.; visualization, D.W. and A.B.; supervision, R.T., K.F.-N. and N.L. All authors have read and agreed to the published version of the manuscript.

**Funding:** This research received no external funding.

**Conflicts of Interest:** The authors declare no conflict of interest.

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
