# Peer review of "Using Historical Responses to Shoreline Change on Australia’s Gold Coast to Estimate Costs of Coastal Adaptation to Sea Level Rise"

_jmse, doi:10.3390/jmse8060380_

Round 1

Reviewer 1 Report

This paper could be  interesting and supply a very good example of sand transfer on the Australian beaches but it is not realistic.

The main problem is that all the paper is based on the Bruun model (1962) which is no more recognized since several years (Cooper & Pilkey, 2004). This model   appears to much simple  to illustrate the  3D transfer processes (along shore and cross shore) of sand on beaches.

The diagram shown on figure 4 does not represent any reality. Authors should present real beach profiles, especially during a long period as it is an historical period. The grain size has to be taking into account and the long linear beach may probably present different profiles in different places. Probably linked to the swell direction.

All other remarks will be pointed line by line.

line 103 : Martinez et al., 2007 : reference number?

line 125 : why only page 146?

line 192 : what kind of document is it? (N°29)

line 224 :( it ) we argue : it should disappear

line 260 : figure 1 : main swell direction should be indicated on the schema.

line 282 : Turner , 2006 reference number?

line 285 : need for a reference for the study of Delpt Hydraulics

Figure 2 : indicate north and south

line 384--389 : you should show these profiles, at least some of them to cover the historical period.

line 391 : Table 1 which should be Table 2: what is the uncertainty of the numbers given in this table ? What does significant the last number? This "precision" does not mean anything and especially compared to the global numbers given for the nourishment in the Table 1 (line 375)

line 399. data are missing : Bathymetry, topography, MNT, swell data, wind rose, tide? grain size?

line 409 : Error bars should be indicated. What are the significative numbers?

line 448 : indicate the units on the top of the table.

After, I am not competent for the economic model and I feel not comfortable to make any remark on this part.

The last point is that for the references, I would really advise the authors to check with paper suggested by  Cooper et Pilkey (2004).

Author Response

Please see the attached file for details of our response to reviewer comments 

Reviewer 2 Report

The paper is a honest research paper. No particular remark , apart some minor editing which is given in the attached file.

In the supplement too, the tables format must be adjusted to avoid numbers to be split on two lines. (not attached because only one file is accepted)

Author Response

Please see our attached file for details of our response to review comments 
